# Cytokines-Biogenesis and Their Role in Human Breast Milk and Determination

**DOI:** 10.3390/ijms22126238

**Published:** 2021-06-09

**Authors:** Anna Kiełbasa, Renata Gadzała-Kopciuch, Bogusław Buszewski

**Affiliations:** 1Department of Environmental Chemistry and Bioanalysis, Faculty of Chemistry, Nicolaus Copernicus University in Toruń, ul. Gagarina 7, 87-100 Toruń, Poland; kielbasam@umk.pl (A.K.); bbusz@umk.pl (B.B.); 2Interdisciplinary Centre of Modern Technologies, Group for Separation and Bioanalytical Methods (Bio-Sep) Nicolaus Copernicus University in Toruń, ul. Wileńska 4, 87-100 Toruń, Poland

**Keywords:** cytokines, ELISA, fatty acids, human breast milk, MALDI, pro- and anti-inflammatory properties

## Abstract

Cytokines play a huge role in many biological processes. Their production, release and interactions are subject to a very complex mechanism. Cytokines are produced by all types of cells, they function very differently and they are characterized by synergism in action, antagonism, and aggregation activity, opposing action of one cytokine, overlapping activity, induction of another cytokine, inhibition of cytokine synthesis at the mRNA level as well as autoregulation-stimulation or inhibition of own production. The predominance of pro-inflammatory cytokines leads to a systemic inflammatory response, and anti-inflammatory-to an anti-inflammatory response. They regulate the organism’s immune response and protect it against sudden disturbances in homeostasis. The synthesis and activity of cytokines are influenced by the central nervous system through the endocrine system (pituitary gland, adrenal glands).

## 1. Introduction

Cytokines are a group of small bioactive proteins, glycoproteins and peptides (Figure 1) that form a complex network of connections between immune cells and modulate the body’s immune response.

Cytokines play important roles during embryonic development, and they can both trigger and inhibit inflammatory responses. Cytokines participate in disease pathogenesis; they are responsible for communication between cells, changes in cognitive functions, senescence processes, responses to infectious and inflammatory factors, specific responses to antigens and infections; and they participate in the differentiation of stem cells. Cytokines are produced mainly by immune system cells (monocytes, macrophages and lymphocytes), neutrophils, B cells and T cells. Cytokines differ in structure and molecular weight (from around 6 to 70 kDa). They can exert additive, synergistic or antagonistic effects, including through mutual induction. Cytokine secretion and concentrations in bodily fluids and tissues are regulated. Cytokines include a wide range of compounds such as chemokines, adipokines, interferons, interleukins, transforming growth factors and tumor necrosis factors [2,3,4].

## 2. Classification of Cytokines

Chemokines (CCL, CXCL) belong to a group of peptides composed of 70 to 130 amino acids. The core domain of a chemokine molecule consists of three β strands stabilized by disulfide bonds and hydrophobic bonds between the β sheet and the α helix. Disulfide bridges between cysteine residues determine the three-dimensional structure of cytokines. Cytokines are produced by leukocytes and tissue cells, and they undergo dimerization and oligomerization. Chemokines are divided into four groups: CXC (alpha), CC (beta), C (gamma) and CX3C (delta), where C is the cysteine residue, and X denotes amino acid residues. Chemokines are chemoattractants that participate in angiogenesis, embryogenesis and organogenesis. Chemokines have pro-inflammatory properties, and they are involved in disease pathogenesis, including pathogenic autoimmune responses and tumor growth (Figure 2) [2,3,4].

Cytokines also include secreted proteins known as interferons (IFNs). Interferons are classified into three types. The largest family of type I INFs (17 proteins) includes IFN-α and IFN-β. Type II INFs contain IFN-γ, and type III INFs include IFN-λ. Interferons are produced immediately after viral infection, and their secretion is stimulated by double-stranded RNA viruses that proliferate in cells [5,6].

Interleukins (IL) are yet another group of cytokines. These small proteins are involved in cell signaling in the immune system. They are produced mainly by leukocytes, fibroblasts, endothelial cells, adipocytes, colonocytes and hematopoietic stem cells. Interleukins are present in epithelial tissues, muscles, skin, blood (IL-1F5, IL-1F6, IL-1F8, IL-1F9, IL-1F10, IL-18, IL-33, IL-36), brain (IL-33, IL-1F5), lungs (IL-33), thymus (IL-36, IL-37, IL-1F5), testicles (IL-36, IL-37), ovaries (IL-36, IL-37), uterus (IL-1F5, IL-36, IL-37), tonsils (IL-33, IL-1F10) and bone marrow (IL-33). Interleukins are generally divided into three groups (Figure 3) based on their biological properties as well as distinguishing structural and molecular features [5,6].

Cytokines also include adipokines which are biologically active substances that are produced exclusively by adipose tissue cells. Adipokines differ considerably in structure and function, and they combine the roles of hormones and cytokines. Adipokines include compounds such as leptin, adiponectin, omentin, resistin, vaspin, visfatin, apelin and chemerin. Adipokines exert direct and indirect effects during apoptosis, angiogenesis, atherogenesis, hemostasis and inflammatory processes, and they regulate blood pressure. Leptin activates the synthesis of pro-inflammatory cytokines TNF-α and IL-6. Adiponectin exhibits antiinflammatory, antisclerotic, antiapoptotic and pro-angiogenic properties, and it enhances insulin sensitivity in tissues. It inhibits the synthesis and activity of TNF-α and the activation of NF-кB. Visfatin regulates insulin secretion. Vaspin demonstrates pro-inflammatory properties and is involved in metabolic disorders. Omentin has pro-inflammatory properties; it enhances insulin sensitivity, stimulates glucose metabolism, and deactivates pro-inflammatory TNF-α and NF-кB. Chemerin is responsible for the secretion of pro-inflammatory cytokines TNF-α and IL-6. Apelin lowers blood pressure and has anti-obesogenic and anti-diabetic effects. Resistin decreases insulin sensitivity [7,8].

Tumor necrosis factor alpha (TNF-α) is a cytokine whose primary structure consists of 157 amino acids. It has two pro-inflammatory forms: a membrane-bound precursor with a molecular weight of 26 kDa, and an enzymatically cleaved form with a molecular weight of 17 kDa. TNF-α exhibits a wide range of biological effects, including immunomodulatory, osteolytic, pyrogenic and pro-inflammatory effects by stimulating the synthesis of IL-1, IL-6, prostaglandins and leukotrienes. Its receptors are found on nearly all nucleated cells. TNF-α is produced by macrophages, monocytes, T cells, natural killer cells, granulocytes, mast cells, endothelial cells, fibroblasts, astrocytes and myocytes. TNF-α is also secreted by certain tumor cells (breast, ovarian, pancreatic tumor cells and melanoma cells). It contributes to insulin resistance by increasing leptin concentration [9,10,11].

Another group of cytokines are colony-stimulating factors (CSF), such as the granulocyte colony-stimulating factor (G-CSF, CSF-3), granulocyte-macrophage colonystimulating factor (GM-CSF, CSF-2) and macrophage colony-stimulating factor (M-CSF, CSF-1). G-CSF is a glycoprotein composed of 174 amino acids. It is responsible for the production and differentiation of neutrophils in the bone marrow. G-CSF is secreted by leukocytes, but it is also produced by some tumor cells. GM-CSF increases the number of circulating blood cells, regulates the immune response and influences leukocyte functions. It is produced by fibroblasts, endothelial cells, T cells, macrophages, mesothelial cells, epithelial cells and various types of tumor cells. CSF-1 is found in bodily fluids and is activated in an autocrine or a paracrine manner. It is responsible for the healthy development of cells, tissues and organs. A deficiency of CSF-1 during development is associated with skeletal, neurological, growth and reproductive abnormalities [12,13,14,15].

Cytokines play a very important role in homeostasis by controlling self-regulatory processes in the body and maintaining stable biological parameters. Seasonal changes in cytokine production are observed in response to environmental factors.

## 3. Cytokine Synthesis in Human Organism, Breast Milk and Their Role in Biogenesis

Cytokines are not only biomarkers of inflammatory diseases, but they can also be used to monitor the effects of drugs on the immune system (Figure 4). Excess production of pro-inflammatory cytokines can lead to tissue damage, hemodynamic changes, organ failure or even death. A cytokine storm occurs when large quantities of pro-inflammatory cytokines are released by the immune system in an uncontrolled manner. Immune cells are suddenly activated, and the immune system loses the ability to control cytokine production. A cytokine storm sets off a chain reaction, where the secreted cytokines induce the production of more cytokines [16,17,18,19,20,21].

A healthy ratio of omega-3 to omega-6 fatty acids plays the key role in the production of pro- and anti-inflammatory cytokines in the human body. These fatty acids and their precursors (exogenous compounds) are not produced by the body and have supplied with the diet. Omega fatty acids are metabolized by the same enzymes, but fatty acids from one omega group cannot be converted to another group [23,24,25,26,27,28,29,30] (Figure 5).

α-Linolenic acid (ALA, the precursor of the omega-3 family), eicosapentaenoic acid (EPA) and docosahexaenoic acid (DHA) are the major fatty acids of the omega-3 family. Fatty acids and their transformation in cells are directly linked with inflammation. Cell membrane phospholipids release polyunsaturated fatty acids which are converted to eicosanoid mediators of inflammation. The cell membrane contains 10–20% arachidonic acid (AA, omega-6), 2–4% DHA (omega-3), and only 0.5–1% of EPA (omega-3). For this reason, AA is usually the dominant precursor for eicosanoid synthesis. EPA and DHA are also metabolized by enzymes, which leads to the synthesis of alternative, competitive compounds of the eicosanoid family [31,32,33] (Figure 6).

Eicosanoids derived from omega-6 fatty acids (AA) are generally classified as proinflammatory, whereas eicosanoids derived from omega-3 fatty acids (EPA and DHA) have anti-inflammatory properties [34].

Eicosanoids are responsible for the production of cytokines which significantly influence pro- and anti-inflammatory processes in the body [31]. Polyunsaturated omega-3 fatty acids inhibit the production of eicosanoids and pro-inflammatory cytokines by decreasing the concentration of AA in cell membranes. Diets that are more abundant in omega-6 than omega-3 fatty acids stimulate the metabolism of omega-6 fatty acids and the production of AA derivatives, and they decrease the metabolic conversion of EPA/DHA to their derivatives. As a result, the concentration of eicosanoids derived from AA increases, which leads to inflammation and the production of pro-inflammatory cytokines. Prostaglandin PGE2 induces the synthesis of pro-inflammatory cytokine IL-6, and leukotriene LTB4 produces pro-inflammatory cytokines such as TNF-α, IL-1 and IL-6. At the same time, the synthesis of anti-inflammatory cytokines decreases. Diets abundant in omega-3 increase the proportions of these fatty acids in cell membrane phospholipids at the expense of AA. The above decreases the synthesis of pro-inflammatory PGE2, thromboxanes TXB2, leukotrienes LTB4 and LTE4, as well as hydroxyeicosatetraenoic acid 5-HETE. E-series resolvins with proven anti-inflammatory properties are derived from EPA. The described enzymes also metabolize DHA to produce anti-inflammatory D-series resolvins, docosatrienes and neuroprotectins. EPA and DHA inhibit the synthesis of pro-inflammatory cytokines TNF-α IL-1β, IL-6 and IL-8. Omega-3 fatty acids probably also change the expression of genes, in particular those encoding pro-inflammatory cytokines [25,31,32,33,34,35] (Figure 7).

Cytokines act on individual cells by binding to specific receptors on their surface. Cytokine receptors are classified into: class I cytokine receptors, class II cytokine receptors, TNF receptors, IL-1 receptors, tyrosine kinase receptors, and chemokine receptors [36].

Cytokines are associated mainly with pro-inflammatory responses, but they do not exert harmful effects on the host organism only. Cytokine activity and cytotoxic and cytostatic effects are determined by cytokine concentrations and the differentiation of target cells. At high concentrations, selected cytokines can inhibit the proliferation of human tumor cell lines, but when their concentrations are low, the same cytokines can stimulate the growth of cancer cells. The application of cytokines with proven anti-inflammatory properties could be a promising approach in preventing and treating inflammations and cancer. However, the thin line between cytokines’ pro- and anti-inflammatory effects could rule out their applicability as potential therapeutic agents.

### 3.1. Cytokines Occurring in Human Breast Milk

Cytokines are present in breast milk, and they are essential for healthy immune system development in newborns. Breast milk is the main source of cytokines, particularly anti-inflammatory cytokines, for newborns that are generally deficient in these proteins. Cytokines have anti-inflammatory effects, and they activate and maintain the body’s immune response. Abnormal cytokine production can have negative health implications, and it can contribute to the development of food allergies, jaundice and immune disorders in later life [16,19]. Pro- and anti-inflammatory cytokines occurring in mothers’ milk are present in Figure 8.

The origin of cytokines in breast milk has not been clearly explained. One of the sources may be breast epithelial cells. Leukocytes such as neutrophils, monocytes/macrophages and lymphocytes migrate to the human breast via the lymphatic vessels and systemic circulation. Leukocytes support the development of the infant’s immune system and fight pathogens directly. It happens as a result of phagocytosis, and the secretion of cytokines and immunoglobulins. These processes occur both in the child’s digestive tract and in tissues to which leukocytes are transferred through systemic circulation of newborn [37].

### 3.2. Impact of Different Factors the Level of Cytokines in Human Breast Milk

Cytokine composition changes in different stages of lactation (Figure 9), and it is also influenced by the mother’s health (diet, depression, stress, allergies) and complications during pregnancy [16,38,39,40,41,42,43].

The largest amount of IL-6 and TNF-α is found in the colostrum due to the changes in the woman’s body during pregnancy and childbirth. The milk of allergic mothers is characterized by a higher concentration of IL-4, IL-13, IL-5, IL-10 and lower TGF-β. Elevated concentrations of IL-6, IL-1β and IL-8 are also associated with inflammations of the mammary gland. At the same time, anti-inflammatory ingredients occur in higher concentrations than in the milk of healthy mothers to protect the infant from developing a clinical disease caused by maternal breastfeeding with mastitis. Higher levels of IL-1β are associated with fatigue in postpartum women. In newborns, in turn, increased IL-1β level is associated with neonatal jaundice. Increased release of cytokines also occurs in the case of sleep disorders. Fatigue, stress and depression increase the risk of infection for both mother and baby. It is assumed that increased IL-1 levels are associated with delivery pain. Depression, stress, or post-traumatic pain can also contribute to increased pro-inflammatory cytokines such as: IL-1β, IL-6, TNF-α, and IFN-γ. Women under stress show increased levels of pro-inflammatory cytokines: IL-6, IL-8 and TNF-α, and the content of anti-inflammatory cytokine IL-10 is much lower. High levels of IL-6 and TNF-α result in preeclampsia and preterm delivery. In the case of the mother’s diet, increased consumption of omega-3 acids reduces the levels of pro-inflammatory cytokines: IL-1α, IL-1β, IL-6 and TNF-α, while the levels of anti-inflammatory cytokines (e.g., IL-10) increase [16,38,39,40,41,42,43,44,45,46,47,48,49].

### 3.3. The Role of Cytokines in Newborns

Exogenous cytokines are very important for the development of the newborn’s organism. They can regulate inflammatory processes, stimulate wound healing, prevent allergies and sepsis, promote hematopoiesis, contribute to healthy gut and thymus development, and increase enterocyte levels.

Interleukin-2 is T lymphocyte regulator and stimulates the growth and development of T lymphocytes and natural killer cells and is involved in the control of Th1/Th2 differentiation. It plays an important role in the development of the immune system, and has a feedback effect on the immune response. After stimulation of the T lymphocyte, it induces molecules on its surface that enable apoptosis of this cell. Pro-inflammatory cytokine IL-6 also affects in the immune system. However, a high concentration of this substance may limit inflammation through a feedback inhibition mechanism. IL-8 is chemoattractant cytokine and pro-inflammatory mediator. It is responsible for recruiting leukocytes and their flow from the mother’s circulation into her milk. Interleukin 4 is important in the process of developing an allergic reaction. It stimulates many different cells of the immune system. It participates in the formation of the inflammatory focus, but the increase in interleukin-4 concentration stimulates hematopoietic processes [50,51,52].

Anti-inflammatory cytokine IL-10 inhibits the development of macrophage cells, T cell, and natural killer cells, but enhances the growth and differentiation of B cells to synthesize immunoglobulins. IL-10 may have immunomodulation and anti-inflammatory effects on the alimentary tract of the newborn. It controls inflammatory processes in the infant organism [51,52].

Pro-inflammatory cytokines such as IL-1β, IL-6, TNF-α, IL-12, IFN-γ, chemokines, and IL-8 can induce systemic inflammation. At the same time, anti-inflammatory cytokines in breast milk such as IL-10 and TGF-β help modulate the cytokine response to infection. This facilitates immune defense and minimizes tissue damage [51,52].

## 4. Methods of Cytokine Detection in Human Breast Milk

Cytokines are detected in various biological matrices with the use of immunoenzymatic (ELISA test, sandwich ELISA, ELISPOT assay, CLIA), histochemical and cytometric methods (CBA-cytometric bead array), as well as techniques that support the quantification of cytokine mRNA (northern blot, PCR). Instrumental techniques which are using to qualification and quantification analysis of cytokines are liquid chromatography, mass spectrometry, matrix assisted laser desorption and ionization (MALDI) as well as surface-enhanced laser desorption/ionization (SELDI). The most popular methods using to determination of cytokine in different matrices are presented on Figure 10.

The enzyme-linked immunosorbent (ELISA) assay detects selected proteins with the use of monoclonal and polyclonal antibodies and enzymes. The modified doble-antibody version of the test is known as sandwich ELISA. The ELISA assay can involve indirect or direct detection methods. In the indirect approach, the antigen is detected with the use of a single enzyme-labeled antibody, whereas the direct approach relies on enzyme-labeled primary and secondary antibodies. The ELISA assay does not provide information about the number or phenotype of cells that produce a given cytokine. This is a quantitative technique where cytokine concentrations are determined based on the calibration curve. The solid-phase enzyme-linked immunosorbent assay (ELISPOT), which involves short-term in vitro culture, is a more sensitive assay than ELISA. The results of the ELISPOT test are influenced by local cytokine concentrations. ELISPOT is also a quantitative technique where cytokines are detected directly on the culture plate. Dark spots on the plate represent cytokine-secreting cells. Cytokines are also detected with the use of immunochemical methods where cell components are identified based on antigen-antibody interactions. The expression of cytokine mRNA is measured in the polymerase chain reaction (PCR) assay or the modified real-time PCR (RTPCR) approach where a specific DNA fragment is multiplied numerous times with the polymerase enzyme. The PCR assay consists of repeated cycles of denaturation of double-stranded DNA, primer annealing and the synthesis of new complementary strands. Cytokine mRNA can also be quantified with the northern-blot technique. This method is generally used to detect mRNA in genes that are transcribed in cells. Cytometric methods are applied to evaluate the size, staining intensity and fluorescence intensity of the analyzed cells [44,53,54,55,56,57,58,59,60,61,62,63,64,65].

Human breast milk is a complex matrix with trace amounts of the examined analytes, and this fact should be taken into account during cytokine detection. Quantitative cytokine analyses should be characterized by high sensitivity and selectivity. The methods for detecting cytokines in human breast milk are presented in Table 1.

In the other matrices chromatographic methods were applied to determine cytokines. An example of such an application is analysis of recombinant granulocyte colony-stimulating factor (rhG-CSF) in pharmaceutical formulations. In this case size-exclusion liquid chromatography with UV detection (214 nm) was used. The mobile phase was phosphoric acid and column a TSK gel G2000 SW column (60 cm × 7.5 mm) [66]. Cytokines can be also determined using mass spectrometry. Such an approach was proposed in an article by Boyle et al. [66], where SELDI/MALDI-TOF-MS was used to determine cytokines (INF-γ, TNF-β, IL-1β, IL-18, IL-6, IL-4) in patient samples. A similar case was described by van Breemen et al. [67]. In that article chemokine CCL-18 was determined in human serum. In human T-cells interleukin-2 (IL-2) can be determined using nanoLC with a hybrid linear ion trap orbitrap mass spectrometer. A Reprosil-Pur C18- Aq (25 cm × 2 cm; 5 μm) column and a gradient mobile phase composed by 0.6% acetic acid and 0.6% acetic acid with 80% ACN:H_2_O were used [68].

## 5. Conclusions

Cytokines have a very complex and broad activity. They work on the principles of pleiotropic, synergism, antagonism, positive and negative feedback. The cytokines found in breast milk are the health source of the newborn baby. These compounds are very important for the development of the newborn’s organism and its immune system. They can regulate inflammatory processes, prevent allergies, are involved in cell apoptosis, induce hematopoietic functions, control inflammatory processes in the developing organism, and minimize tissue damage.

Cytokines are responsible for induction of and a controlled course immune response. There is a known phenomenon called “cytokine storm”. It is characterized by an excessive and uncontrolled reaction of the immune system that releases huge amounts of pro-inflammatory cytokines. There is a sudden activation of cells of the immune system, that loses the ability to control the inhibition of the excessive production of cytokines. A chain reaction occurs in which the released cytokines induce the formation of others. It is an unstoppable process. 

Different cytokines can have the same biological effects. Many studies show that pro-inflammatory cytokines are actively involved in carcinogenesis. There is a close correlation between the amount of unsaturated fatty acids in the human body and the synthesis of anti- and pro-inflammatory cytokines. 

## Figures and Tables

**Figure 1 ijms-22-06238-f001:**
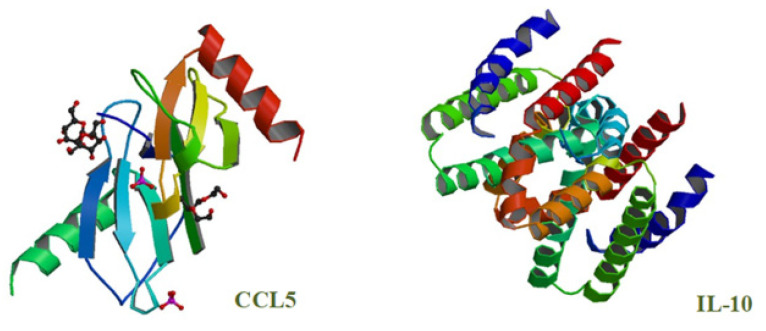
Structure of selected cytokines: chemokines CCL5 and interleukins IL-10 (source: [1]).

**Figure 2 ijms-22-06238-f002:**
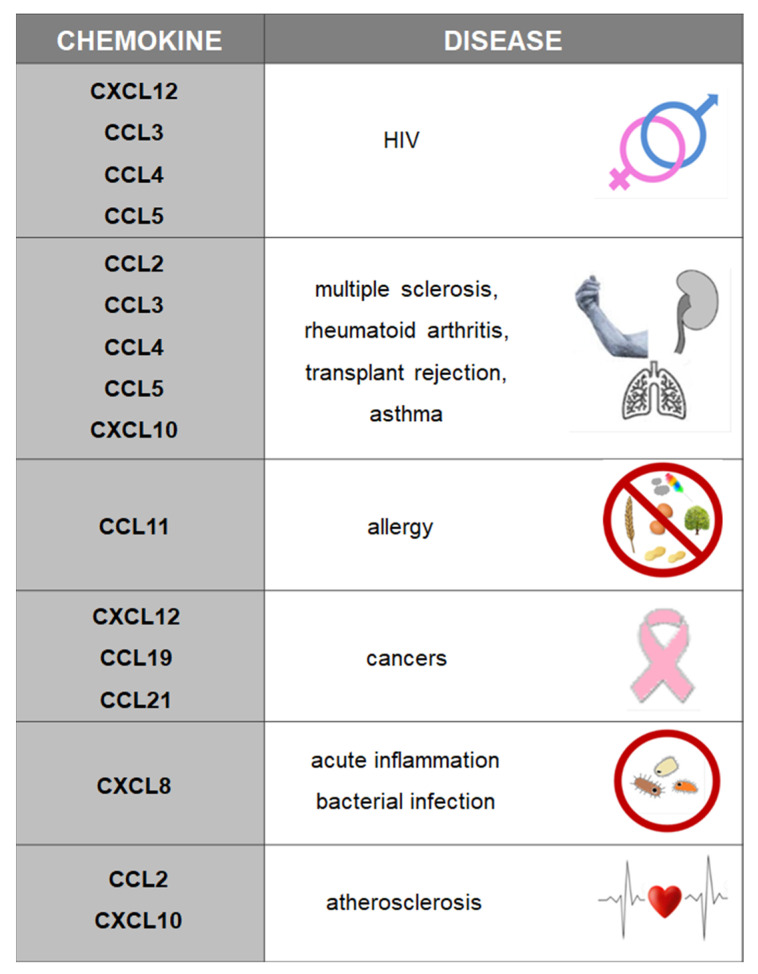
Chemokines and the associated diseases.

**Figure 3 ijms-22-06238-f003:**
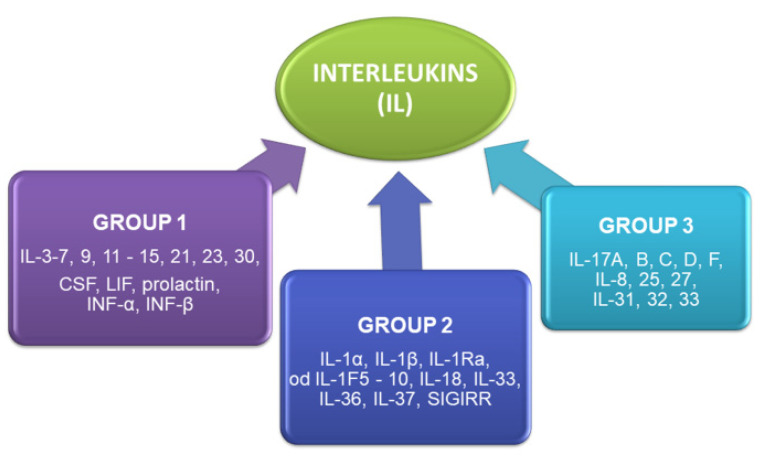
Major interleukin groups.

**Figure 4 ijms-22-06238-f004:**
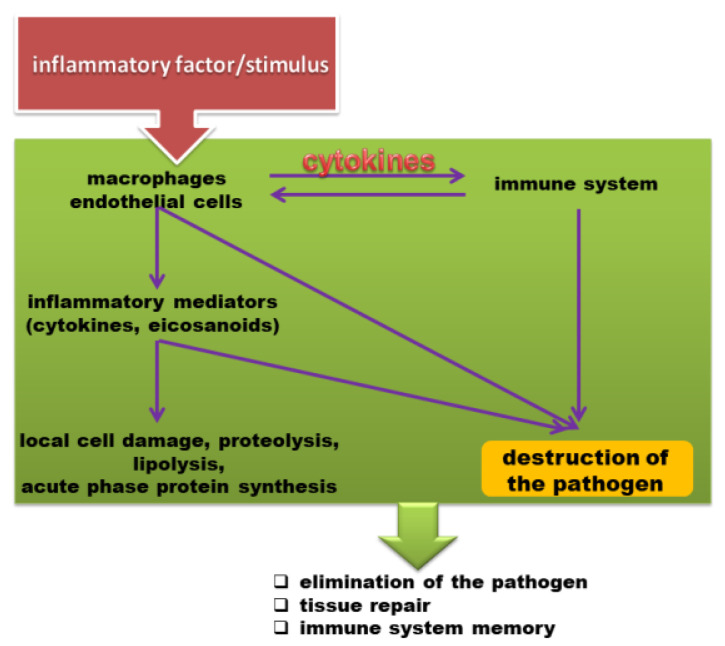
Specific immune response [22].

**Figure 5 ijms-22-06238-f005:**
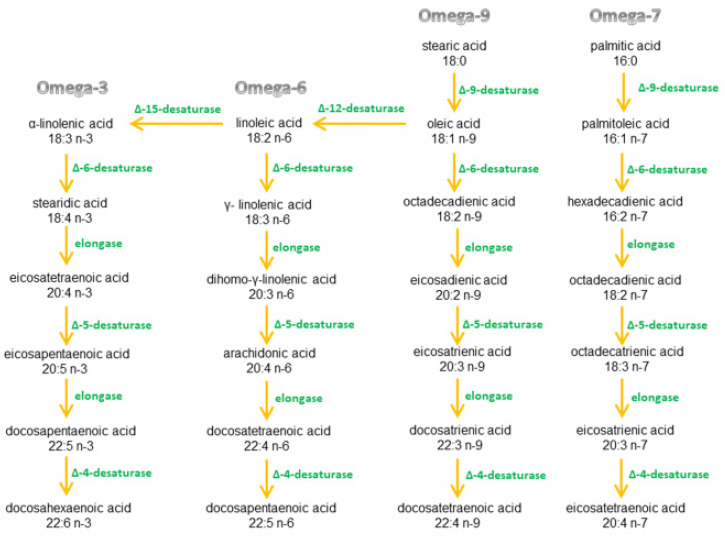
Fatty acid metabolism.

**Figure 6 ijms-22-06238-f006:**
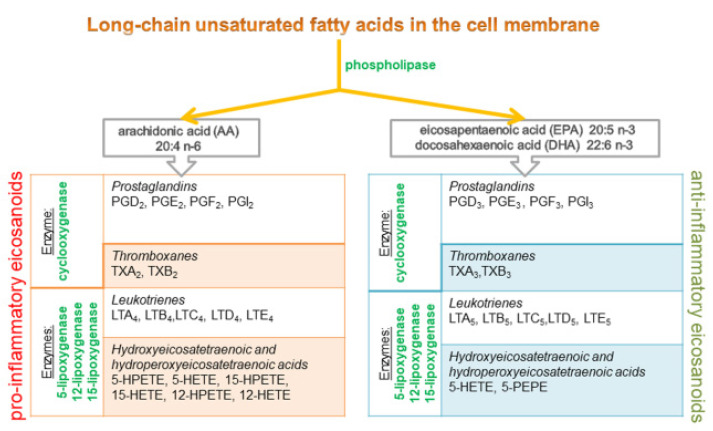
Two pathways of eicosanoid synthesis from fatty acids.

**Figure 7 ijms-22-06238-f007:**
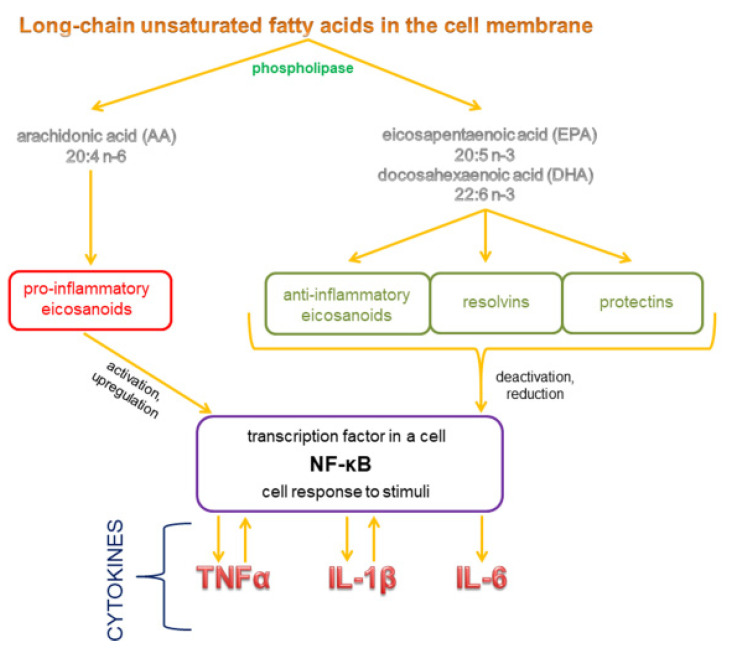
The mechanism by which fatty acids influence cytokine synthesis.

**Figure 8 ijms-22-06238-f008:**
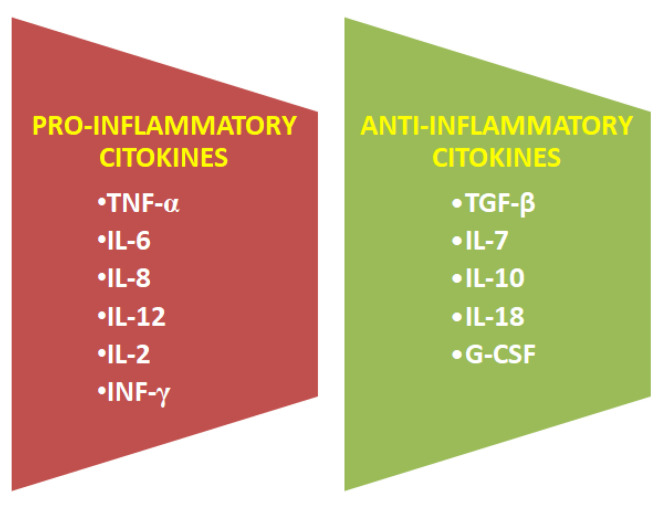
Cytokines present in human breast milk.

**Figure 9 ijms-22-06238-f009:**
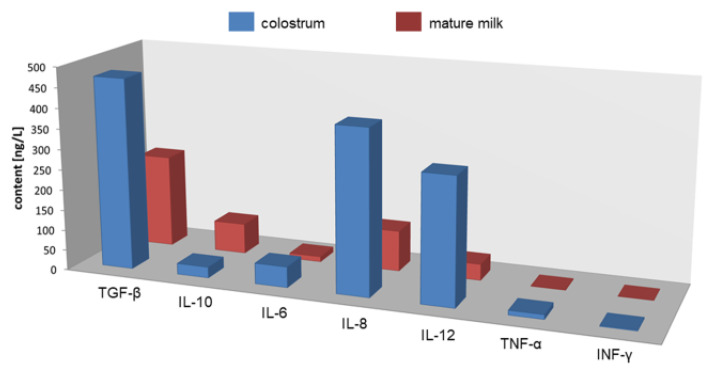
Cytokine concentrations in breast milk in different stages of lactation.

**Figure 10 ijms-22-06238-f010:**
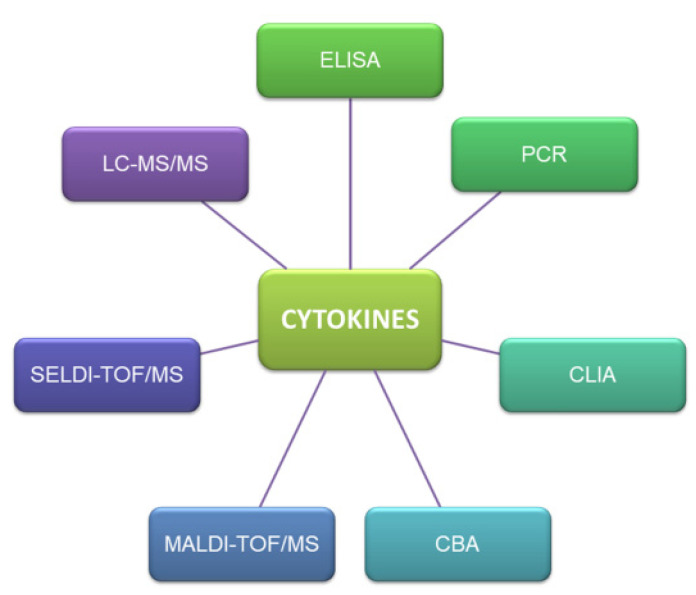
The popular methods of cytokines analysis.

**Table 1 ijms-22-06238-t001:** Methods of cytokine detection in human breast milk.

Cytokine	Concentration	Matrix	Method	Reference
IL-1α	0–3791 pg/mL		Multiplex kits containing polystyrene beads coated with antibodies	[53]
IL-1β	0–212.1 pg/mL			
	5.0–266.0 pg/mL	Colostrum	Chemiluminescence enzyme immunometric assay (CLIA) in an Immulite Immunoassay Analyzer	[44]
	6.6–140.0 pg/mL	Preeclampsia-colostrum		
	5.0 pg/mL	Regular milk		
	5.0–316.0 pg/mL	Preeclampsia-regular milk		
	4.0 pg/mL	Control colostrum	Chemiluminescence enzyme immunometric assay (CLIA) in an Immulite Immunoassay Analyzer	[43]
	8.7 pg/mL	Colostrum-neonatal jaundice		
TGF-β	177–471 pg/mL	Mature milk from non-allergic mothers	ELISA (enzyme-linked immunosorbent assay) on high-adsorption polystyrene microtitration plates (NUNC)	[54]
	189–432 pg/mL	Mature milk from allergic mothers		
	0–4700 pg/mL	Colostrum from non-allergic mothers	ELISA (enzyme-linked immunosorbent assay) with continuous plate shaking during incubation	[55]
	0–3400 pg/mL	Colostrum from allergic mothers		
	0–6250 pg/mL	Mature milk from non-allergic mothers		
	0–2400 pg/mL	Mature milk from allergic mothers		
TGF-β_1_	1200 pg/mL	Colostrum form non-allergic mothers	ELISA (enzyme-linked immunosorbent assay) with continuous plate shaking during incubation	[55]
	330 pg/mL	Colostrum from allergic mothers		
	1059 pg/mL	Mature milk from non-allergic mothers		
	215 pg/mL	Mature milk from allergic mothers		
	0.13–2.40 µg/L	Colostrum produced after full-term pregnancy	Enzyme-linked immunosorbent assay (ELISA)	[56]
	0.25–15.8 µg/L	Colostrum produced after preterm delivery		
	0.28–1.41 µg/L	Colostrum produced after extremely preterm delivery		
	0.13–0.72 µg/L	Mature milk produced after full-term pregnancy		
	0.13–1.16 µg/L	Mature milk produced after preterm delivery		
	0.16–0.71 µg/L	Mature milk produced after extremely preterm delivery		
	152–1165 pg/mL	Colostrum	Enzyme-linked immunosorbent assay (ELISA)	[57]
	125–616 pg/mL	Mature milk		
	67–186 pg/mL	Colostrum	Commercial sandwich ELISA: Latent TGF-β_1_ is activated during incubation at room temperature for 1 h with 1 mol/L HCl and neutralized with an equal volume of 1.2 mol/L NaOH/0.5 mol/L HEPES	[61]
	17–114 pg/mL	Mature milk		
	28–3542 pg/mL		Enzyme-linked immunosorbent assay (ELISA)	[62]
	73–1768 pg/mL	Colostrum	Enzyme-linked immunosorbent assay (ELISA)	[63]
TGF-β_2_	0.09–13.3 µg/L	Colostrum produced after full-term pregnancy	Enzyme-linked immunosorbent assay (ELISA)	[56]
	1.43–43.0 µg/L	Colostrum produced after preterm delivery		
	1.26–4.82 µg/L	Colostrum produced after extremely preterm delivery		
	0.10–12.8 µg/L	Mature milk produced after full-term pregnancy		
	0.20–6.24 µg/L	Mature milk produced after preterm delivery		
	0.31–6.17 µg/L	Mature milk produced after extremely preterm delivery		
	357–16,784 pg/mL	Colostrum	Enzyme-linked immunosorbent assay (ELISA)	[57]
	250–11,696 pg/mL	Mature milk		
	1376–5394 pg/mL	Colostrum	Commercial sandwich ELISA; latent TGF-β_2_ activation: incubation at room temperature for 1 h with 1 mol/L acetic acid and neutralization with 2 volumes of 1.2 mol/L NaOH/0.5 mol/L HEPES	[61]
	592–2697 pg/mL	Mature milk		
	98 to 13,855 pg/mL		Enzyme-linked immunosorbent assay (ELISA)	[62]
	39–2240 pg/mL	Colostrum	Enzyme-linked immunosorbent assay (ELISA)	[63]
	153–42,117 ng/L	Milk from allergic mothers		
	255–21,094 ng/L	Milk from non-allergic mothers	Enzyme-linked immunosorbent assay (ELISA)	[38]
TNF-α	0.35–36.9 ng/L	Colostrum produced after full-term pregnancy	Cytometric bead array (CBA) human soluble protein flex set	[56]
	3.0–51.1 ng/L	Colostrum produced after preterm delivery		
	0.35–11.0 ng/L	Colostrum produced after extremely preterm delivery		
	0.35–5.45 ng/L	Mature milk produced after full-term pregnancy		
	0.35–8.6 ng/L	Mature milk produced after preterm delivery		
	0.35–3.1 ng/L	Mature milk produced after extremely preterm delivery		
	5–38 pg/mL	Milk produced after preterm delivery	Enzyme-linked immunosorbent assay (ELISA)	[58]
	5–20 pg/mL	Milk produced after full-term pregnancy		
	0–68.8 pg/mL		Multiplex kits containing polystyrene beads coated with antibodies	[53]
	14.0–253.0 pg/mL	Colostrum	Chemiluminescence enzyme immunometric assays (CLIA) in an Immulite Immunoassay Analyzer	[44]
	26.0–172.0 pg/mL	Preeclampsia-colostrum		
	4.0–13.3 pg/mL	Regular milk		
	7.2–69.0 pg/mL	Preeclampsia-regular milk		
	21.9 pg/mL	Colostrum	Chemiluminescence enzyme immunometric assays (CLIA) in an Immulite Immunoassay Analyzer	[43]
	28.6 pg/mL	Colostrum-neonatal jaundice		
	<4.4–15.1 ng/L	Milk from non-allergic mothers	Enzyme-linked immunosorbent assay (ELISA)	[38]
	<4.4–21.3 ng/L	Milk from allergic mothers		
IFN-γ	49–708 pg/mL	Milk from non-allergic mothers	ELISA (enzyme-linked immunosorbent assay) on high-adsorption polystyrene microtitration plates (NUNC)	[54]
	14–240 pg/mL	Milk from allergic mothers		
	67–113 pg/mL	Colostrum	ELISA (enzyme-linked immunosorbent assay) with continuous plate shaking during incubation	[57]
	67–69 pg/mL	Mature milk		
	2–35 pg/mL	Milk produced after preterm delivery	Enzyme-linked immunosorbent assay (ELISA)	[58]
	2–20 pg/mL	Milk produced after full-term pregnancy		
	0–276.9 pg/mL		Multiplex kits containing polystyrene beads coated with antibodies	[53]
IL-2	0–321.7 pg/mL		Multiplex kits containing polystyrene beads coated with antibodies	[53]
IL-4	61–172 pg/mL	Milk from non-allergic mothers	ELISA method on high-adsorption polystyrene microtitration plates (NUNC)	[54]
	82–362 pg/mL	Milk from allergic mothers		
	5.6–91 pg/mL	Colostrum	ELISA kits with continuous plate shaking during incubation	[57]
	5.6–109 pg/mL	Mature milk		
	0–3714.4 pg/mL		Multiplex kits containing polystyrene beads coated with antibodies	[53]
IL-5	70–189 pg/mL	Milk from non-allergic mothers	ELISA on high-adsorption polystyrene microtitration plates (NUNC)	[54]
	76–126 pg/mL	Milk from allergic mothers		
	6.2–11 pg/mL		Analyzed on high-binding, half-area Costar 3690 plates, coated with 50 μL/well of 0.25 μg/mL monoclonal rat anti-human IL-5	[57]
	0–111.7 pg/mL		Multiplex kits containing polystyrene beads coated with antibodies	[53]
IL-6	51–105 pg/mL	Milk from non-allergic mothers	ELISA on high-adsorption polystyrene microtitration plates (NUNC)	[54]
	37–111 pg/mL	Milk from allergic mothers		
	4.4–340 ng/L	Colostrum produced after full-term pregnancy	Cytometric bead array (CBA) human soluble protein flex set	[56]
	15.3–362 ng/L	Colostrum produced after preterm delivery		
	9.3–67.9 ng/L	Colostrum produced after extremely preterm delivery		
	1.6–26.4 ng/L	Mature milk produced after full-term pregnancy		
	2.2–226 ng/L	Mature milk produced after preterm delivery		
	2.6–53.3 ng/L	Mature milk produced after extremely preterm delivery		
	5.6–279 pg/mL	Colostrum	ELISA kits with continuous plate shaking during incubation	[57]
	5.6–247 pg/mL	Mature milk		
	0–1173.9 pg/mL		Multiplex kits containing polystyrene beads coated with antibodies	[53]
	31.8–528.0 pg/mL	Colostrum	Chemiluminescence enzyme immunometric assays (CLIA) in an Immulite Immunoassay Analyzer	[44]
	5.0–309.0 pg/mL	Preeclampsia-colostrum		
	5.0–9.0 pg/mL	Regular milk		
	5.0–572.0 pg/mL	Preeclampsia-regular milk		
	<5–49 pg/mL	Colostrum	Enzyme-linked immunosorbent assay (ELISA)	[63]
	5.1 pg/mL	Milk from healthy mothers	Protein microarray method	[64]
	2.4 pg/mL	Milk from allergic mothers		
	15.5 pg/mL	Colostrum	Chemiluminescence enzyme immunometric assays (CLIA) in an Immulite Immunoassay Analyzer	[43]
	17.5 pg/mL	colostrum with neonatal jaundice		
IL-7	0–1071.3 pg/mL		Multiplex kits containing polystyrene beads coated with antibodies	[53]
IL-8	0.04–26.3 µg/L	Colostrum produced after full-term pregnancy	Cytometric bead array (CBA) human soluble protein flex set	[56]
	0.13–14.7 µg/L	Colostrum produced after preterm delivery		
	0.13–2.98 µg/L	Colostrum produced after extremely preterm delivery		
	0.01–0.41 µg/L	Mature milk produced after full-term pregnancy		
	0.01–0.22 µg/L	Mature milk produced after preterm delivery		
	0.02–0.23 µg/L	Mature milk produced after extremely preterm delivery		
	0–9795.2 pg/mL		Multiplex kits containing polystyrene beads coated with antibodies	[53]
	1079–14,300 pg/mL	Colostrum	Chemiluminescence enzyme immunometric assays (CLIA) in an Immulite Immunoassay Analyzer	[44]
	3957–14,250 pg/mL	Preeclampsia-colostrum		
	65–236 pg/mL	Regular milk		
	73–14,500 pg/mL	Preeclampsia-regular milk		
	82.5 pg/mL	Milk from healthy mothers	Protein microarray method	[64]
	102 pg/mL	Milk from allergic mothers		
	583.0 pg/mL	Colostrum	Chemiluminescence enzyme immunometric assays (CLIA) in an Immulite Immunoassay Analyzer	[43]
	751.5 pg/mL	Colostrum-neonatal jaundice		
IL-10	27–429 pg/mL	Milk from healthy mothers	ELISA on high-adsorption polystyrene microtitration plates (NUNC)	[54]
	27–400 pg/mL	Milk from allergic mothers		
	4.8 pg/mL	Colostrum from allergic mothers	ELISA with continuous plate shaking during incubation	[55]
	9.5 pg/mL	Mature milk from allergic mothers		
	0.06–31.1 ng/L	Colostrum produced after full-term pregnancy	Cytometric bead array (CBA) human soluble protein flex set	[56]
	0.06–23.4 ng/L	Colostrum produced after preterm delivery		
	0.06–7.8 ng/L	Colostrum produced after extremely preterm delivery		
	0.06–4.5 ng/L	Mature milk produced after full-term pregnancy		
	0.06–2.5 ng/L	Mature milk produced after preterm delivery		
	0.06–4.24 ng/L	Mature milk produced after extremely preterm delivery		
	19–202 pg/mL	Colostrum	ELISA kits with continuous plate shaking during incubation	[57]
	19–139 pg/mL	Mature milk		
	<5–25.72 pg/mL	Colostrum milk	Solid-phase sandwich enzyme-linked immunosorbent assay kit	[59]
	<5–17.21 pg/mL	Transitional milk		
	<5 pg/mL	Mature milk		
	0–1898.7 pg/mL		Multiplex kits containing polystyrene beads coated with antibodies	[53]
	0–42 pg/mL	Colostrum and mature milk from allergic and non-allergic mothers	ELISA according to the manufacturer’s instructions with continuous plate shaking during incubation	[55]
	<5–604 pg/mL	Colostrum	Enzyme-linked immunosorbent assay (ELISA)	[63]
	6.5 pg/mL	Colostrum	Chemiluminescence enzyme immunometric assays (CLIA) in an IMMULITE Immunoassay Analyzer	[43]
	4.3 pg/mL	Colostrum-neonatal jaundice		
	<3.9–10.8 ng/L	Milk from allergic mothers	Enzyme-linked immunosorbent assay (ELISA)	[38]
	<3.9–49.4 ng/L	Milk from non-allergic mothers		
IL-12	<3–25.12 pg/mL	Colostrum	Solid-phase sandwich enzyme-linked immunosorbent assay kit	[59]
	<3–10.18 pg/mL	Transitional milk		
	<3 pg/mL	Mature milk		
	<40->10,000 pg/mL	Colostrum	Enzyme-linked immunosorbent assay (ELISA)	[60]
	<40–3935 pg/mL	Transitional milk		
	<40–1648 pg/mL	Mature milk		
	3–24 pg/mL	Mature milk produced after full-term pregnancy	Enzyme-linked immunosorbent assay (ELISA)	[58]
	3–33 pg/mL	Mature milk produced after preterm delivery		
	0–305.4 pg/mL		Multiplex kits containing polystyrene beads coated with antibodies	[53]
	<40–10,000 pg/mL	Colostrum	Enzyme-linked immunosorbent assay (ELISA)	[60]
	<40–3935 pg/mL	Transitional milk		
	<40–1648 pg/mL	mature milk		
IL-13	151–270 pg/mL	Milk from healthy mothers	ELISA method on high-adsorption polystyrene microtitration plates (NUNC)	[54]
	93–1359 pg/mL	Milk from allergic mothers		
	3.2–89 pg/mL	Colostrum	ELISA with continuous plate shaking during incubation	[57]
	3.2–56 pg/mL	Mature milk		
	0–724.3 pg/mL		Multiplex kits containing polystyrene beads coated with antibodies	[53]
IL-15	0–470 pg/mL		Multiplex kits containing polystyrene beads coated with antibodies	[53]
IL-17	0–87.2 pg/mL		Multiplex kits containing polystyrene beads coated with antibodies	[53]
IL-18	<12.5–875 pg/mL	Colostrum	Enzyme-linked immunosorbent assay (ELISA)	[58]
	<12.5–1375 pg/mL	Transitional milk		
	<12.5–250 pg/mL	Mature milk		
GM-CSF	0–325 pg/mL		Multiplex kits containing polystyrene beads coated with antibodies	[53]

## Data Availability

Not applicable.

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
