# Peer review of "Cytokines-Biogenesis and Their Role in Human Breast Milk and Determination"

_ijms, 2021, doi:10.3390/ijms22126238_

Round 1

Reviewer 1 Report

Thank you for asking me to review this manuscript which addresses cytokines and their role in human breast milk.

Minor comments:

  1. Chapter 3 should be amended. Please write 3 separate subsections:

3.1-which of the described cytokines are present in breast milk (general description),

3.2-what factors affect the level of cytokines in breast milk (which factors increase or decrease the levels of different cytokines),

3.3-what may be the effects of different levels of cytokines in newborns.

  1. Please shorten the description for eicosanoids.
  2. Please correct the Conclusions (should apply to subsection 3.3).

Thank you

Author Response

Dear Reviewer;

We would like to thank the first Reviewer for valuable comments regarding the submitted manuscript. All corrections are in green color.

  1. Chapter 3 should be amended. Please write 3 separate subsections:

3.1-which of the described cytokines are present in breast milk (general description),

3.2-what factors affect the level of cytokines in breast milk (which factors increase or decrease the levels of different cytokines),

3.3-what may be the effects of different levels of cytokines in newborns.

Answer: Appropriate corrections are included in the manuscript.

  1. Please shorten the description for eicosanoids.

Answer: Appropriate corrections are included in the manuscript. The second of the Reviewers accepted this chapter, so we are afraid that more interference in its content could result in the lack of acceptance for this article.

  1. Please correct the Conclusions (should apply to subsection 3.3).

Answer: Appropriate corrections are included in the manuscript.

Best regards,

Renata Gadzała-Kopciuch

Reviewer 2 Report

This review appears interesting, though with some criticism to be further addressed.The very long list of data about cytokine levels is pleonastic and does not deal with the complex immune milieu of breastfeeding mothers' milk. Cells are dismissed in this review, an are very important to be introduced in the Review to explain how cytokines are.

Author Response

Dear Reviewer; 

We would like to thank the first Reviewer for valuable comments regarding the submitted manuscript.  

By answering the reviewer's question, we agree with the Reviewer in his comments. However, our review is on the verge of biochemistry, and we do not dare to enter the competence of excellent scientists specializing in cell biology and immunology. We do not feel competent enough to interfere in this way with the article.

Best ragards,

Renata Gadzała-Kopciuch

Round 2

Reviewer 2 Report

The authors have fullfilled the Reviewer's comments